# Metabolomics approach for predicting stomach and colon contents in dead *Arctocephalus pusillus pusillus, Arctocephalus tropicalis, Lobodon carcinophaga* and *Ommatophoca rossii* from sub-Antarctic region

**Mukhethwa Micheal Mphephu[1]☯, Oyinlola Oluwunmi Olaokun[1‡], Caswell Mavimbela[1‡], Greg Hofmeyer[2‡], Monica Mwale[3‡], Nqobile Monate Mkolo[1]☯***

1 Department of Biology, School of Science and Technology, Sefako Makgatho Health Science University, Ga-Rankuwa, Pretoria, South Africa, 2 Port Elizabeth Museum at Bayworld, Humewood, Port Elizabeth, South Africa, 3 South African National Biodiversity Institute (SANBI), National Zoological Garden, Pretoria, South Africa

☯ These authors contributed equally to this work.
‡ These authors also contributed equally to this work.
* Nqobile.mkolo@smu.ac.za

## Abstract

The dietary habits of seals play a pivotal role in shaping management and administration policies, especially in regions with potential interactions with fisheries. Previous studies have utilized various methods, including traditional approaches, to predict seal diets by retrieving indigestible prey parts, such as calcified structures, from intestines, feces, and stomach contents. Additionally, methods evaluating nitrogen and stable isotopes of carbon have been employed. The metabolomics approach, capable of quantifying small-scale molecules in biofluids, holds promise for specifying dietary exposures and estimating disease risk. This study aimed to assess the diet composition of five seal species—*Arctocephalus pusillus pusillus, Lobodon carcinophaga, Ommatophoca rossii*, and *Arctocephalus tropicalis* 1 and 2—by analyzing stomach and colon contents collected from stranded dead seals at various locations. Metabolite concentrations in the seal stomach and colon contents were determined using Nuclear Magnetic Resonance Spectroscopy. Among the colon and stomach contents, 29 known and 8 unknown metabolites were identified. Four metabolites (alanine, fumarate, lactate, and proline) from stomach contents and one metabolite (alanine) from colon contents showed no significant differences between seal species (p>0.05). This suggests that traces of these metabolites in the stomach and colon contents may be produced by the seals' gut microbiome or derived from other animals, possibly indicating reliance on fish caught at sea. Despite this insight, the cause of death for stranded seals remains unclear. The study highlights the need for specific and reliable biomarkers to precisely indicate dietary exposures across seal populations. Additionally, there is a call for the development of relevant metabolite and disease interaction networks to explore disease-related metabolites in seals. Ultimately, the

**Data Availability Statement:** All relevant data are within the manuscript and its Supporting Information files.

**Funding:** Applicant: Mr MUKHETHWA MPHEPHU Grant numbers: MND200602526966 Institution: Sefako Makgatho Health Sciences University Funder: NRF Postgraduate Scholarships No the the sponsors or funders did not play any role in the study design, data collection and analysis, decision to publish, or preparation of the manuscript.

**Competing interests:** The authors have declared that no competing interests exist.

metabolomic method employed in this study reveals potential metabolites in the stomach and colon contents of these seal species.

## Introduction

Marine mammals are secondarily aquatic vertebrates, and in particular, the pinnipeds constitute a monophyletic clade of marine carnivorans that originated from terrestrial ancestors approximately 27 million years ago [1,2]. The transition from terrestrial to aquatic habitats presented marine mammals with new physiological challenges. One crucial adaptation for the survival of these marine mammal lineages was the development of the ability to capture and consume prey in the marine environment [3].

Mammals foraging in water must navigate changes in density, pressure, and viscosity in the aquatic environment. Additionally, depending on the depth, they must contend with reduced visibility. These environmental pressures have led to ecological and morphological specializations related to food processing, locomotion, and prey capture [3]. Pinnipeds, which include seals, maintain a dual terrestrial–aquatic lifestyle, utilizing land for thermoregulation, breeding, and molting. However, their foraging activities take place entirely underwater, driving the evolution of specialized sensory adaptations for prey detection, including eyesight, hearing, and specialized vibrissae [3]. Notably, these sensory systems are also crucial for their terrestrial activities.

Ross seals (*Ommatophoca rossii*) and crabeater seals (*Lobodon carcinophaga*) are phocid seals (earless seals) with a circumpolar distribution in the Southern Ocean (Antarctica) [4,5]. While crabeater seals are the most abundant pinniped species globally, Ross seals are the rarest, most elusive, pelagic and Antarctic ice-inhabiting species [4–6]. Ross seals feed on a more varied diet of squid, crustaceans and fish, while the crabeater seals feed only on specialized diet of the small crustacean, the Antarctic krill (*Euphausia superba*) [7]. With a regional study suggesting that seals eat more Antarctic krill than is taken by the krill fishery [8]. The fur seals of the genus *Arctocephalus* such as the sub-Antarctic fur seal (*Arctocephalus tropicalis*) and Cape fur seal (*Arctocephalus pusillus pusillus*), inhabit a wide range of marine regions, including sub-Antarctic and tropical waters respectively [9]. Sub-Antarctic fur seal is widespread in the Atlantic, Indian, and Pacific Oceans and primarily feed on fishes, mainly myctophids, and small amounts of cephalopods and crustaceans [10]. The Cape fur seal is endemic to the African continent and has a diet that includes teleost fish, elasmobranchs, cephalopods, crustaceans, and seabirds [11]. As two-thirds of the Cape fur seal's diet comprise commercially targeted species, there is ongoing concern about resource competition between the seal population, and commercial fisheries [10].

Thus, understanding the foraging ecology of seals, including trophic interactions, feeding behavior, and the factors influencing them, is crucial for comprehending ecosystem functioning and potential impacts on ecosystems due to changes in seal populations [12]. This knowledge is essential for developing management and administration policies, especially where there is a potential link with fisheries. Worldwide, increasing seal populations have led to resource competition and conflicts with commercial fisheries [12]. Balancing the conservation importance of seals with the economic costs of seal–fishery interactions is an ongoing challenge.

Direct studies of seal feeding are logistically challenging due to protection regulations, deep-sea foraging habits, and remote habitats [3]. The diet of seal species, including *A. pusillus pusillus*, *L. carcinophaga*, *O. rossii*, and *A. tropicalis*, has been previously studied [13–17] using

**Table 1. The basic morphometrics parameters of the different stranded and deceased seals species.**

| Accession Number | Species | Age class | Sex | Location collected | Date collected |
|---|---|---|---|---|---|
| PEM N5284 | *Arctocephalus pusillus pusillus* | subadult | unknown | Maitland, Eastern Cape | 21-Jan-17 |
| PEM N5453 1 | *Arctocephalus tropicalis* | adult | female | Maitland, Eastern Cape | 24-Aug-17 |
| PEM N5433 2 | *Arctocephalus tropicalis* | yearling | female | Plettenberg Bay, Western Cape | 14-Feb-18 |
| PEM N5455 | *Lobodon carcinophaga* | under yearling | female | Jeffrey's Bay, Eastern Cape | 23-Jan-17 |
| PEM N5458 | *Ommatophoca rossii* | adult | male | Princess Martha Coast, Antarctica | 14-Jan-16 |

methods such as the analysis of prey indigestible parts retrieved from intestines, feces, and stomach contents [18,19]. Less frequently used methods involve examining spewing for prey indigestible parts [20,21] or employing chemical techniques such as quantitative fatty acid signature analysis [22], examination of prey DNA retrieved from stomachs and feces of seals [23,24], and evaluation of nitrogen and stable isotopes of carbon [25,26]. While these methods provide valuable insights, the integration of a metabolomics approach to study the stomach and colon contents from stranded dead seals of *A. pusillus pusillus*, *L. carcinophaga*, *O. rossii*, and *A. tropicalis* species has not been explored. Metabolomics can quantify small-scale molecules in biofluids, offering a more specific understanding of dietary exposures and potential disease risks. This approach may account for metabolic irregularities influenced by various factors, including environmental and genetic factors. By measuring downstream components or metabolic products of foods, metabolomics can provide insights into exposure to non-nutritive substances, such as toxic chemicals, which may be crucial for understanding disease etiology [27].

Hence, this study utilized a metabolomics approach to analyze the stomach and colon contents of stranded deceased seals belonging to various species, such as *A. pusillus pusillus*, *L. carcinophaga*, *O. rossii*, and *A. tropicalis*. The objective of this approach was to advance our understanding of the identified metabolites, investigate their associations with metabolites related to diseases in both animals and humans, and provide insights into potential dietary exposures and possibly the cause of death of the seals.

## Materials and methods

### Sample collection and ethics statement

The basic morphometric parameters of *A. pusillus pusillus*, *L. carcinophaga*, *O. rossii*, and two *A. tropicalis* species, such as age class and sex, are presented in Table 1. Stomach and colon contents were collected from stranded deceased seals of *A. pusillus pusillus*, *L. carcinophaga*, *O. rossii*, and *A. tropicalis* 1 and 2 species at various locations: Maitland (Eastern Cape), Jeffrey's Bay (Eastern Cape), Princess Martha Coast (Antarctica), Plettenberg Bay (Western Cape), and Maitland (Eastern Cape), respectively. These samples, including stomach, colon, and whole animal specimens, were collected by G. Hofmeyer from Bayworld and subsequently stored at the National Zoological Garden (NZG) forensic laboratory Pretoria branch at -80˚C. Collection permits for stomach and colon samples were obtained from the Department of Agriculture, Land Reform, and Rural Development, South Africa, under section 20 of animal diseases (12/11/1/1/18; 12/11/1/12 (1745JD), and the SMUREC ethics certificate (SMUREC/S/17/2020:PG).

### Sample preparation

Each sample of the seal stomach and colon contents was divided for two Nuclear Magnetic Resonance spectroscopy (NMR) analyses, involving 1) analysis of the colon material and 2) analysis of the stomach material. To each sample, 500 μL of ddH2O was added, followed by vortexing and centrifugation. For the first analysis (colon material), 540 μL of the supernatant

was transferred to a microcentrifuge tube containing 60 µL of NMR buffer solution (1.5 M potassium phosphate solution in deuterium oxide with the internal standard TSP—trimethyl-silyl-2,2,3,3-tetradeuteropropionic acid [0.5805 mM]; pH 7.4). The sample suspension was then vortexed and centrifuged at 12,000 x g for 5 minutes to eliminate any solid particulates that might have remained. The final volume of 550 µL of the resulting supernatant was transferred to a 5 mm NMR tube for analytical analyses. For the second analysis (stomach material), 1 mL of ddH2O was added to each of the original sample pellets. A 3 mm tungsten carbide bead was then added to each sample vial, and the mixture was subsequently ground and homogenized in a Mixer Mill MM 400 for 3 minutes at 30 Hz to disrupt the pellet. Afterward, 540 µL of each homogenized sample was transferred to a microcentrifuge tube containing 60 µL of NMR buffer solution (1.5 M potassium phosphate solution in deuterium oxide with the internal standard TSP; pH 7.4). The sample suspension was then vortexed and centrifuged at 12,000 x g for 5 minutes to remove any solid particulates that might have remained. The final volume of 550 µL of the resulting supernatant was then transferred to a 5 mm NMR tube for analytical analyses.

## H-NMR analyses

Samples were analyzed at 500 MHz using a Bruker Avance III HD NMR spectrometer equipped with a triple resonance inversus (TXI) $^{1}$H [$^{15}$N, $^{13}$C] probe head and x, y, z gradient coils. For $^{1}$H spectra, 128 transients were acquired in 32K data points, with a spectral width of 10,504 Hz and an acquisition time of 3.12 s. The receiver was set to 64. The sample temperature was maintained at 300 K, and the $H_2O$ resonance was pre-saturated by single-frequency irradiation during a relaxation delay of 4 s, with a 90˚ excitation pulse of 8 µs. Automatic shimming of the sample was performed on the deuterium signal. Fourier transformation, phase correction, and baseline correction were carried out automatically using Bruker Topspin software (V3.5).

## Data processing

Spectra were first calibrated in automation to the TSP signal (0.00 pm). Relative metabolite concentrations were calculated by fitting the reference signals in the raw spectra with specific Voigt functions. To account for the potential dilution effect, each metabolite set was subsequently normalized using Probabilistic Quotient Normalization (PQN). Missing metabolite measurements were imputed to zero value.

## Bioinformatic and statistics analysis

One-way analysis of variance (ANOVA) was conducted to determine the differential expression and adjusted p-value (False Discovery Rate) cutoff of 0.05 among the various species, including *A. pusillus pusillus*, *L. carcinophaga*, *O. rossii*, and *A. tropicalis* 1 and 2, regarding their colon and stomach contents. The correlation between the colon and stomach contents of these species was assessed using NCSS 2021, v21.0.3. Correlations were computed within and between the species of *A. pusillus pusillus*, *L. carcinophaga*, *O. rossii*, and *A. tropicalis* 1 and 2. The detection of metabolites with higher or lower intensity was achieved by applying a filtering criterion of at least 2.0-fold for the colon and stomach contents of the different species. MetaboAnalyst 5.0 (http://www.metaboanalyst.ca/faces/home.xhtml) was employed for the comprehensive analysis of metabolomic data. MetaboAnalyst 5.0 was utilized to generate hierarchical cluster analysis for creating a heatmap of the produced metabolites. Additionally, MetaboAnalyst 5.0 was employed for the visualization of significant metabolites enriched in specific pathways. The tool was further utilized for network analysis in three distinct manners:

metabolite pathway analysis, debiased sparse partial correlation (DSPC) metabolite network, and metabolite and disease interaction network.

## Results

### Two-dimensional hierarchical clustering of metabolic profiling

A total of 29 known and 8 unknown metabolites (S1 File) were identified within the colon and stomach contents. Among these metabolites, six expressed from stomach contents (alanine, fumarate, lactate, proline, sarcosine, and urocanate) showed no significant differences between the seal groups ($p > 0.05$). Additionally, only alanine expressed from the colon contents exhibited no significant differences between the seal groups ($p > 0.05$) (Fig 1). The segregation of metabolites into groups was visualized using a heatmap with hierarchical clustering (Figs 2A and 3B). Figs 2B and 3B heatmaps also depict and summarize the correlation of metabolites among different seal species. Furthermore, irrespective of the seal species (*A. pusillus pusillus*, *L. carcinophaga*, *O. rossii*, and *A. tropicalis* 1 and 2), there was a strong correlation (0.902, 0.999, 0.989, 0.904, and 0.972) between metabolites from the colon and stomach contents, respectively. Fig 4 provides a summary of the correlation between species of *A. pusillus pusillus*, *L. carcinophaga*, *O. rossii*, and *A. tropicalis* 1 and 2.

Furthermore, in the comparisons of stomach content metabolites between *A. tropicalis* 1 and *O. rossii*, *A. tropicalis* 1 and *L. carcinophaga*, *A. tropicalis* 1 and *A. pusillus pusillus*, and *A. tropicalis* 1 and *A. tropicalis* 2, a total of 10, 14, 17, and 5 metabolites were upregulated with a fold change >2, respectively (Fig 5). In the comparison between *O. rossii* and *L. carcinophaga*,

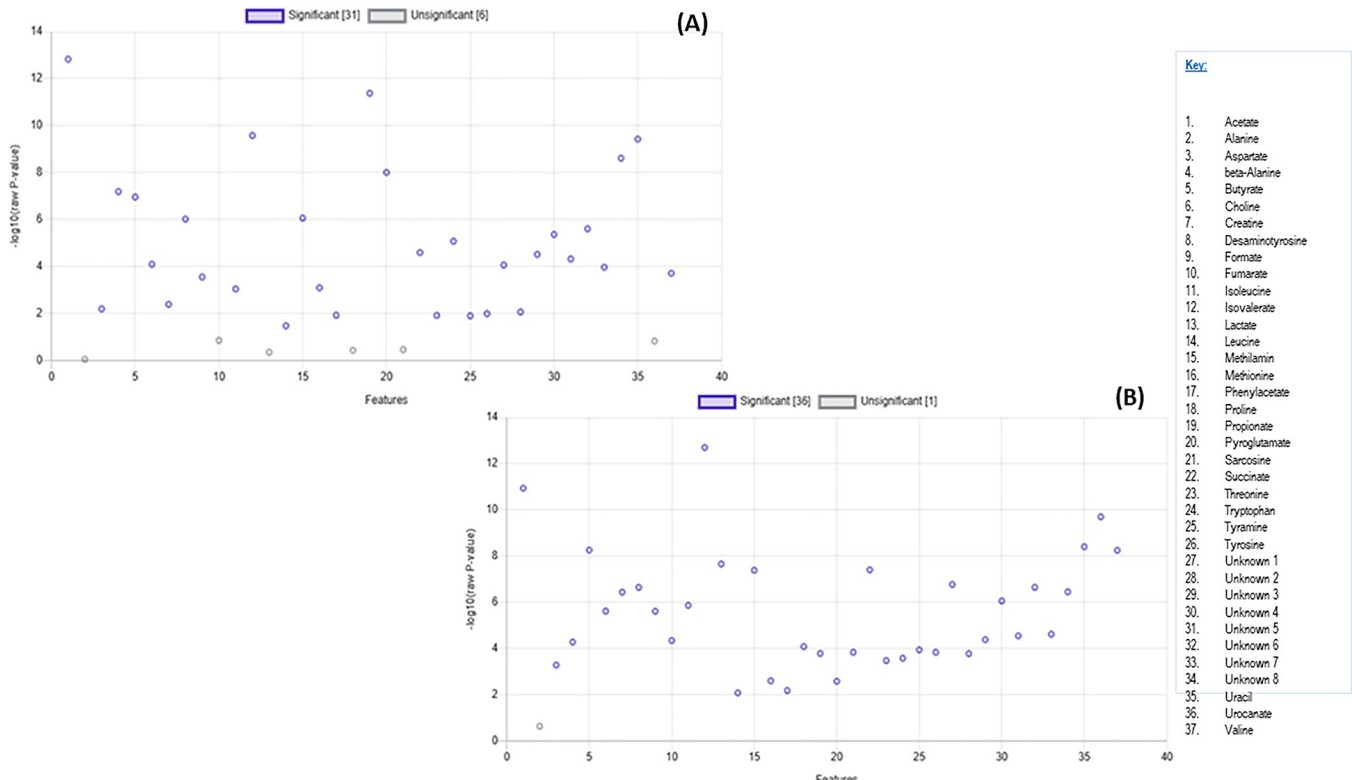

**Fig 1.** A scatter plot illustrating the 37 metabolites that are insignificant and significant based on one-way ANOVA from the (A) stomach content and (B) colon content of different seals of *A. pusillus pusillus*, *L. carcinophaga*, *O. rossii*, *A. tropicalis* 1 and *2* species.

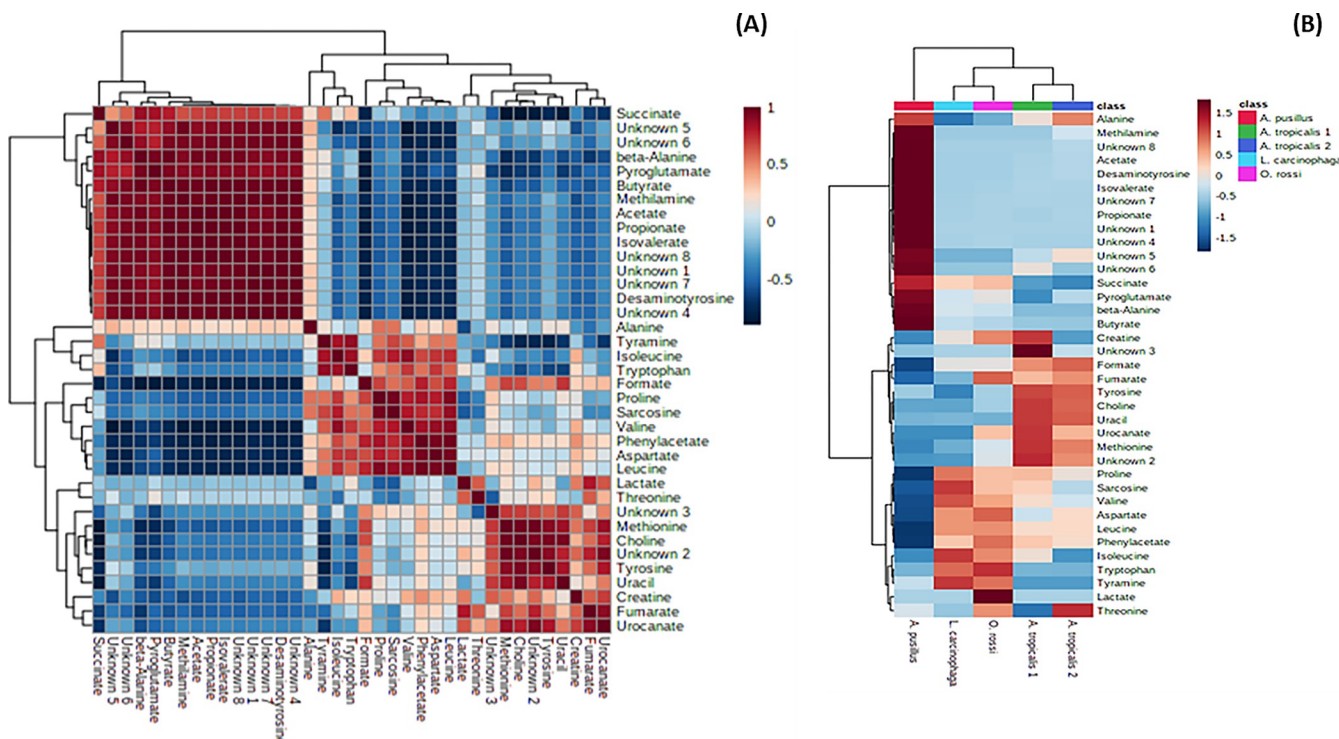

**Fig 2. A heatmap illustrating the 37 metabolites clearly segregated in samples of *A. pusillus pusillus*, *L. carcinophaga*, *O. rossii* and *A. tropicalis* 1 and 2 species for stomach contents.** (A) Each colored cell on the map represents the correlation value of metabolites in the data table. (B) Each colored cell on the map corresponds to a concentration value in the data table, where samples are in columns and metabolites are in rows. The heatmap was employed to identify metabolites with high or low concentrations.

no metabolites from the stomach contents were upregulated. However, 14 and 7 metabolites were upregulated with a fold change greater than 2 when *O. rossii* metabolites were compared to those of *A. pusillus pusillus* and *A. tropicalis* 2 metabolites, respectively (Fig 6). Additionally, in the comparisons of stomach content metabolites between *L. carcinophaga*, and *A. pusillus pusillus*, *L. carcinophaga*, and *A. tropicalis* 2, and *A. pusillus pusillus* and *A. tropicalis* 2, a total number of 16, 15, and 16 metabolites were upregulated with a fold change greater than 2, respectively (Fig 7).

In the comparison of colon content metabolites between *A. tropicalis* 1 and *O. rossii*, *A. tropicalis* 1 and *L. carcinophaga*, *A. tropicalis* 1 and *A. pusillus pusillus*, and *A. tropicalis* 1 and *A. tropicalis* 2, a total of 10, 12, 16, and 8 metabolites were upregulated with a fold change greater than 2, respectively (Fig 8). Additionally, in the comparison between *O. rossii* and *L. carcinophaga*, metabolites, *O. rossii* and *A. pusillus pusillus* metabolites, and *O. rossii* and *A. tropicalis* 2 metabolites, a total of 11, 16, and 16 metabolites from the colon contents were upregulated with a fold change greater than 2, respectively (Fig 9). Furthermore, in the comparison of colon content metabolites between *L. carcinophaga*, and *A. pusillus pusillus*, *L. carcinophaga*, and *A. tropicalis* 2, and *A. pusillus pusillus* and *A. tropicalis* 2, a total number of 16, 15, and 15 metabolites were upregulated with a fold change greater than 2, respectively (Fig 10).

## Metabolite pathway analysis

Pathway analysis was conducted utilizing both enrichment and topology analyses. The enrichment analysis revealed metabolites significantly enriched in specific biochemical pathways

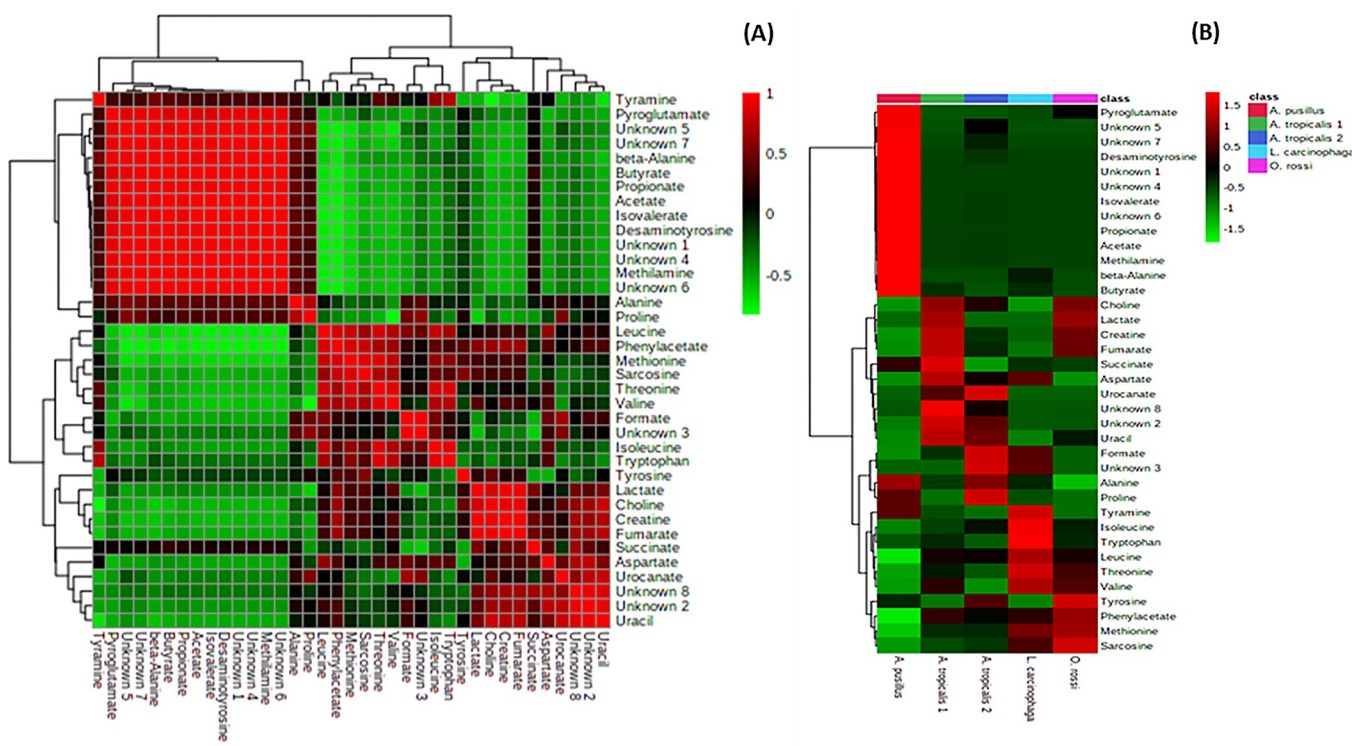

**Fig 3. A heatmap illustrating the 37 metabolites clearly segregated in samples of *A. pusillus pusillus*, *L. carcinophaga*, *O. rossii* and *A. tropicalis* 1 and 2 species for colon contents.** (A) Each colored cell on the map represents a correlation value of metabolites in the data table. (B) Each colored cell on the map corresponds to a concentration value in the data table, with samples in columns and metabolites in rows. The heatmap was employed to identify metabolites with high or low concentrations.

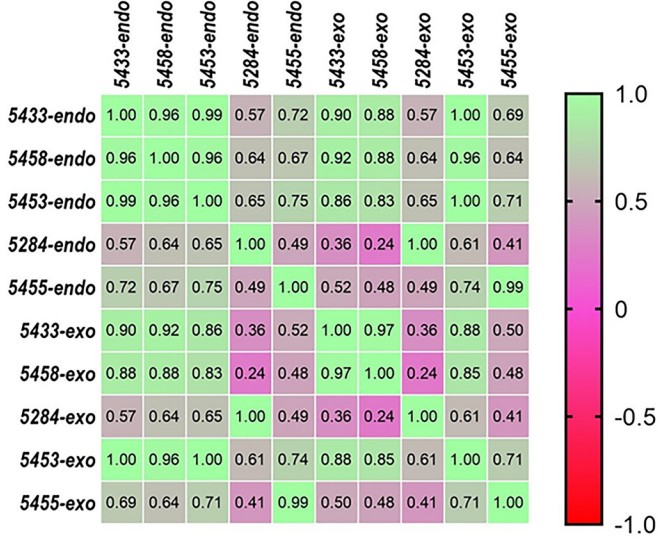

| | 5433-endo | 5458-endo | 5453-endo | 5284-endo | 5455-endo | 5433-exo | 5458-exo | 5284-exo | 5453-exo | 5455-exo |
|---|---|---|---|---|---|---|---|---|---|---|
| **5433-endo** | 1.00 | 0.96 | 0.99 | 0.57 | 0.72 | 0.90 | 0.88 | 0.57 | 1.00 | 0.69 |
| **5458-endo** | 0.96 | 1.00 | 0.96 | 0.64 | 0.67 | 0.92 | 0.88 | 0.64 | 0.96 | 0.64 |
| **5453-endo** | 0.99 | 0.96 | 1.00 | 0.65 | 0.75 | 0.86 | 0.83 | 0.65 | 1.00 | 0.71 |
| **5284-endo** | 0.57 | 0.64 | 0.65 | 1.00 | 0.49 | 0.36 | 0.24 | 1.00 | 0.61 | 0.41 |
| **5455-endo** | 0.72 | 0.67 | 0.75 | 0.49 | 1.00 | 0.52 | 0.48 | 0.49 | 0.74 | 0.99 |
| **5433-exo** | 0.90 | 0.92 | 0.86 | 0.36 | 0.52 | 1.00 | 0.97 | 0.36 | 0.88 | 0.50 |
| **5458-exo** | 0.88 | 0.88 | 0.83 | 0.24 | 0.48 | 0.97 | 1.00 | 0.24 | 0.85 | 0.48 |
| **5284-exo** | 0.57 | 0.64 | 0.65 | 1.00 | 0.49 | 0.36 | 0.24 | 1.00 | 0.61 | 0.41 |
| **5453-exo** | 1.00 | 0.96 | 1.00 | 0.61 | 0.74 | 0.88 | 0.85 | 0.61 | 1.00 | 0.71 |
| **5455-exo** | 0.69 | 0.64 | 0.71 | 0.41 | 0.99 | 0.50 | 0.48 | 0.41 | 0.71 | 1.00 |

Key:

| Stomach samples | Colon samples | Species name |
|---|---|---|
| 5284-endo | 5284-endo | *Arctocephalus pusillus* |
| 5433-endo | 5433-endo | *Arctocephalus tropicalis 1* |
| 5453-endo | 5453-endo | *Arctocephalus tropicalis 2* |
| 5455-endo | 5455-endo | *Lobodon carcinophaga* |
| 5458-endo | 5458-endo | *Ommatophoca rossii* |

**Fig 4. Summaries the correlation of stomach content between species of *A. pusillus pusillus*, *L. carcinophaga*, *O. rossii* and *A. tropicalis* 1 and 2.**

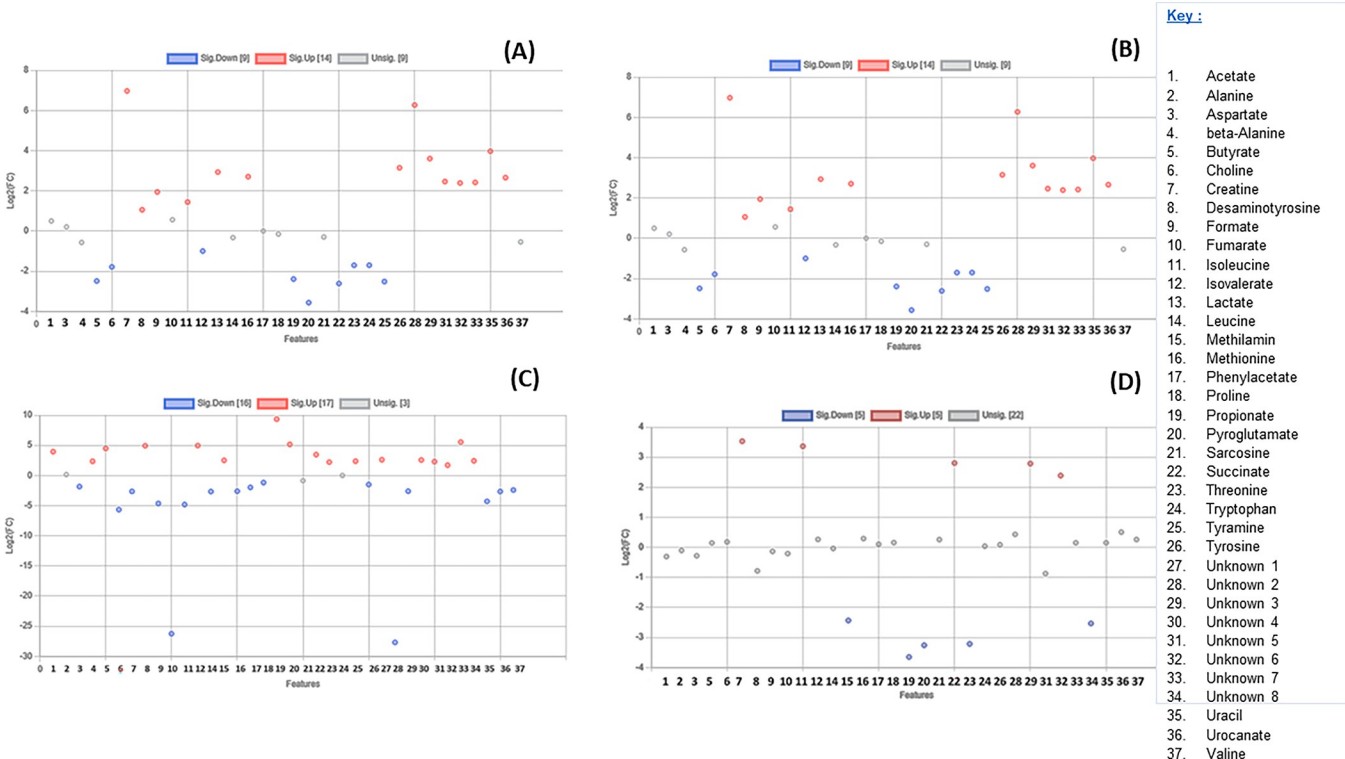

**Fig 5. A scatter plot illustrating the metabolites that are insignificant and significantly downregulated and upregulated in the stomachs of different seals.** (A) Metabolites in *A. tropicalis* 1 versus *O. rossii*. (B) Metabolites in *A. tropicalis* 1 versus *L. carcinophaga*. (C) Metabolites in *A. tropicalis* 1 versus *A. pusillus pusillus* (D) Metabolites in *A. tropicalis* 1 versus *A. tropicalis* 2.

($P < 0.05$; Fig 11), encompassing aminoacyl-tRNA biosynthesis, valine, leucine, and isoleucine biosynthesis, glycine, serine, and threonine metabolism, alanine, aspartate, and glutamate metabolism, Isoquinoline alkaloid biosynthesis, tyrosine metabolism, β-alanine metabolism, pantothenate and CoA biosynthesis, sulfur metabolism, valine, leucine, and isoleucine degradation, β-alanine metabolism, pyruvate metabolism, propanoate metabolism, and phenylalanine metabolism. The topology analysis identified metabolites likely playing crucial roles in pathways based on their positions within these pathways. In the comparison between the colon and stomach contents of *A. pusillus pusillus*, *L. carcinophaga*, *O. rossii*, and *A. tropicalis* 1 and 2, the consistently perturbed pathways included Aminoacyl-tRNA biosynthesis, valine, leucine, and isoleucine biosynthesis, and glycine, serine, and threonine metabolism.

## Debiased Sparse Partial Correlation (DSPC) metabolite network

The metabolite interaction network visually highlights correlations between functionally related metabolites. Commonly mapped correlations among functionally related metabolites of *A. pusillus pusillus*, *L. carcinophaga*, *O. rossii*, and *A. tropicalis* 1 and 2 were observed. Metabolites identified through metabolomics were integrated into the Debiased Sparse Partial Correlation (DSPC) network to construct a subnetwork (Fig 12A–12E).

The subnetwork comprises 36 nodes (metabolites), four of which were downregulated, and 147 edges. The nodes in the subnetwork also represent the most significantly enriched pathways, including aminoacyl-tRNA biosynthesis, valine, leucine, and isoleucine biosynthesis, glycine, serine, and threonine metabolism, pantothenate and CoA biosynthesis, and alanine, aspartate, and glutamate metabolism (Table 2, Fig 12A–12E).

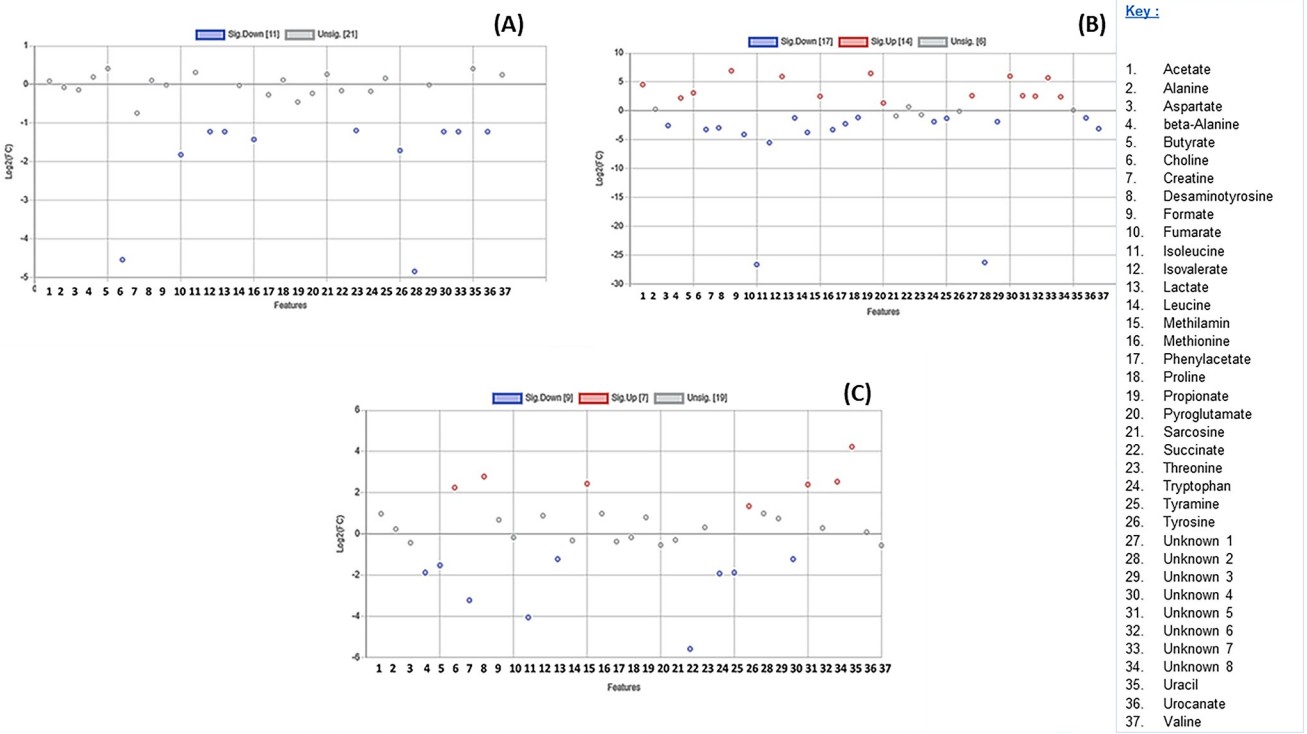

**Fig 6. A scatter plot illustrating the metabolites that are insignificant and significantly downregulated and upregulated in the stomachs of different seals.** (A) Metabolites in *O. rossii* versus *L. carcinophaga*, (B) Metabolites in *O. rossii* versus *A. pusillus pusillus* (C) Metabolites in *O. rossii* versus *A. tropicalis* 2.

## Metabolite and disease interaction network

The metabolite and disease interaction network provide insights into disease-related metabolites. Metabolites identified through metabolomics were integrated into the metabolite and disease (animals and humans) interaction network, resulting in the creation of two subnetworks (Fig 13). subnetwork 1 consists of 119 nodes, 162 edges, and 21 seed metabolites and diseases. subnetwork 2 comprises 3 nodes, 2 edges, and 1 seed metabolite and disease. Alzheimer's disease emerges as the most significantly associated disease with different metabolites (Fig 13).

## Discussion

This study represents the first application of metabolomics to elucidate potential molecular signaling pathways and networks associated with the dietary exposure and potential causes of death in seals. Conducting an untargeted global analysis of urine and fecal samples proves to be a valuable approach in identifying metabolite biomarkers linked to diet or disease [28]. Our investigation focused on the metabolic activity within the stomach and colon ecosystems of stranded seals species. A total of 29 known and 8 unknown metabolites, encompassing amino acids, organic compounds, fatty acids, enzymes, and other microbial-origin metabolites, were extracted. Regardless of the seal species a significant correlation was observed between metabolites in the colon and stomach contents. However, minimal to no correlation was noted between different seal species, which might be attributed to the distinct dietary preferences of seals for certain prey species [29]. Specifically, six metabolites expressed from stomach contents (alanine, fumarate, lactate, proline, sarcosine, and urocanate) exhibited no statistical significance (p>0.05) among the seal species. Moreover, only alanine expressed from the colon

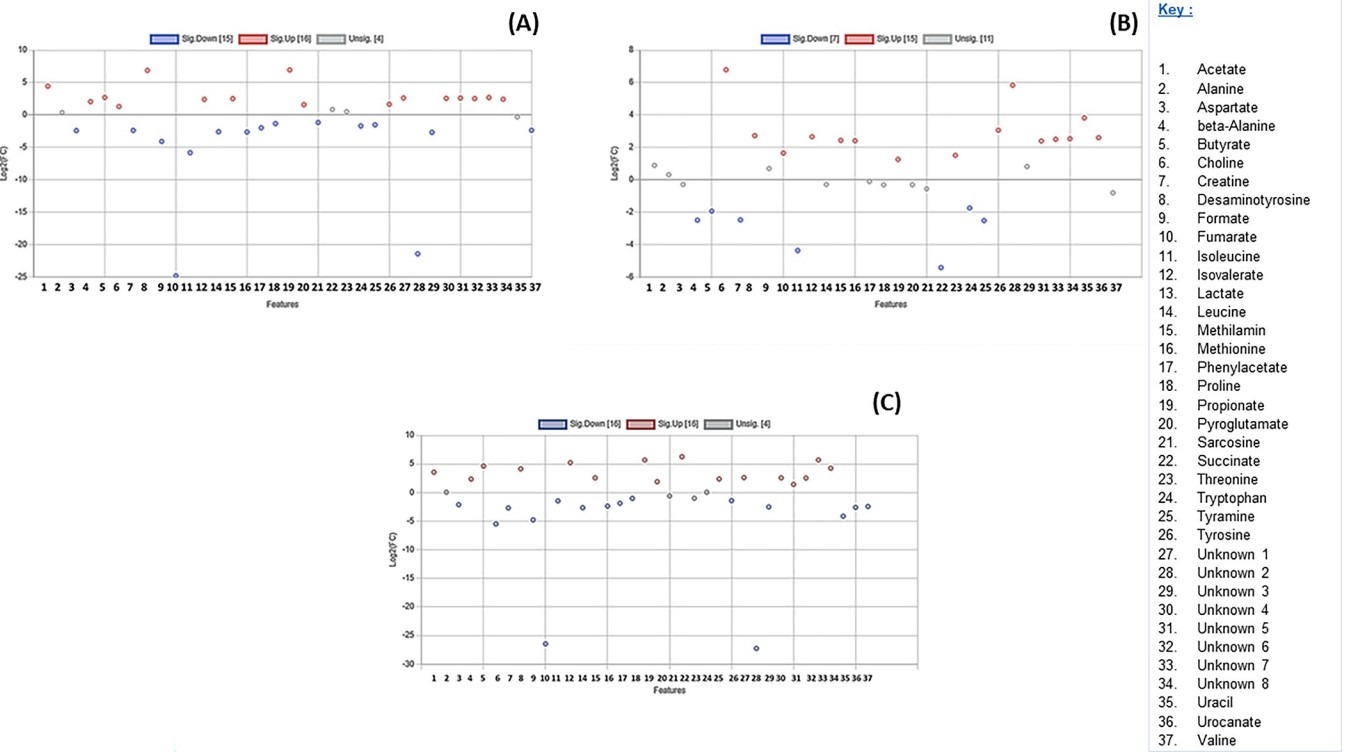

**Fig 7. A scatter plot illustrating the metabolites that are insignificant and significantly downregulated and upregulated in the stomachs of different seals.** (A) Metabolites *in L. carcinophaga*, versus *A. pusillus pusillus* (B) Metabolites in *L. carcinophaga*, versus *A. tropicalis* 2. (C) Metabolites in *A. pusillus pusillus* versus *A. tropicalis* 2.

contents among the seal species was statistically insignificant (p>0.05). These findings imply that the extracted metabolites were dependent on the seal species, with 31 and 36 differentially expressed metabolites identified in the stomach and colon, respectively.

Furthermore, twenty metabolites (acetate, alanine, aspartate, butyrate, choline, desamino-tyrosine, formate, fumarate, isoleucine, phenylacetate, proline, propionate, pyroglutamate, sarcosine, succinate, threonine, tryptophan, tyrosine, unknown 2, uracil) were consistently found in the stomach contents of all seal species. Similarly, 16 metabolites (acetate, alanine, butyrate, choline, creatine, formate, fumarate, isoleucine, phenylacetate, propionate, pyroglutamate, sarcosine, succinate, tryptophan, unknown 2, and uracil) were identified in the colon contents of all seal species. The presence of these shared metabolites enhances our understanding of the similarities in the metabolic profiles across different seal species.

To further understand the significance of the metabolites identified through the NMR method, pathway analysis was performed. The functional analysis of metabolites from the colon and stomach contents of the seals revealed their involvement in different pathways. Aminoacyl-tRNA biosynthesis (succinate, sarcosine, propionate, threonine, acetate, urocanate, uracil, tryptophan, isovalerate, pyroglutamate) exhibited the highest significance. Metabolites associated with valine, leucine and isoleucine biosynthesis included uracil, propionate, isovalerate, and acetate. Alanine, aspartate, and glutamate metabolism were associated with metabolites such as desaminotyrosine, pyroglutamate, creatine, and succinate. Glycine, serine, and threonine metabolism were linked to uracil, lactate, aspartate, and β-alanine.

Aminoacyl-tRNAs are crucial in protein synthesis and have been implicated in various physiological and pathological processes beyond translation [30]. Some diseases, such as

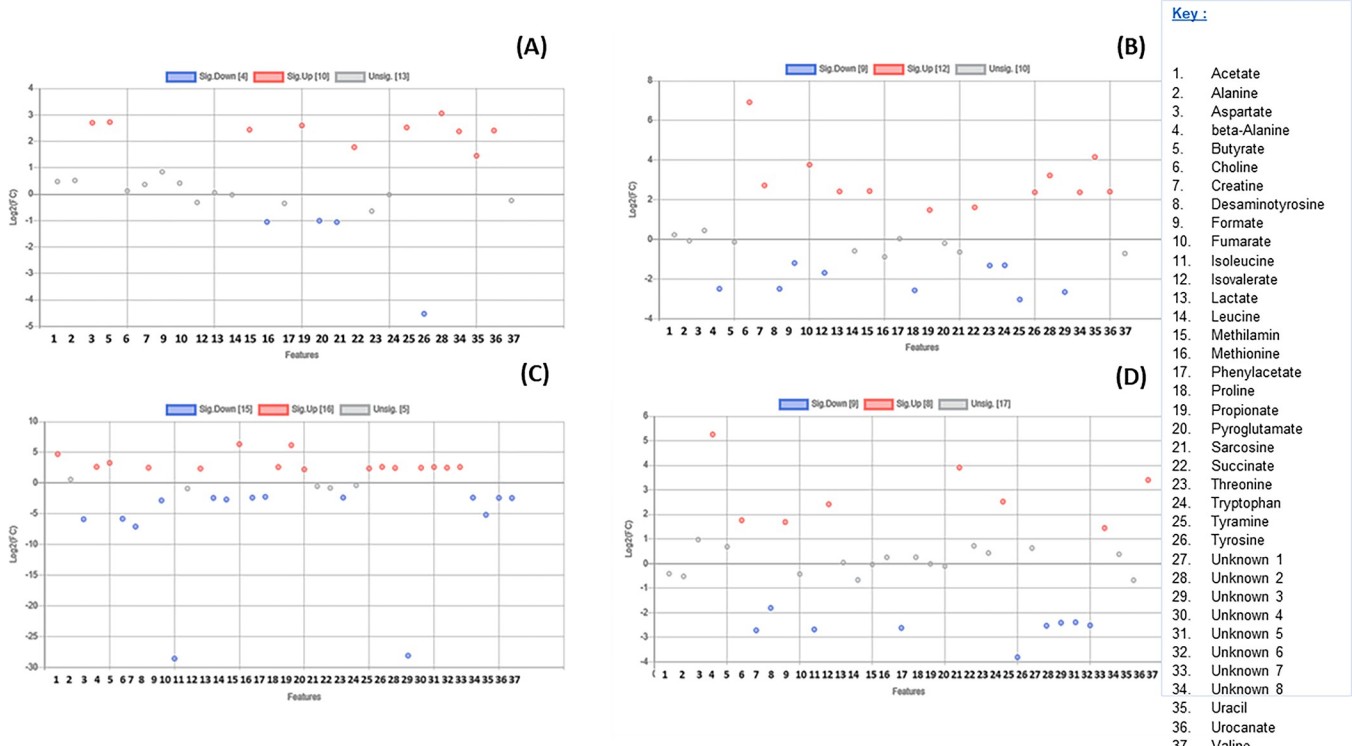

**Fig 8. A scatter plot illustrating the metabolites that are insignificant and significantly downregulated and upregulated in the colon of different seals.** (A) Metabolites in *A. tropicalis* 1 versus *O. rossii*. (B) Metabolites in *A. tropicalis* 1 versus *L. carcinophaga*, (C) Metabolites in *A. tropicalis* 1 versus *A. pusillus pusillus* (D) Metabolites in *A. tropicalis* 1 versus *A. tropicalis* 2.

neuronal pathologies, autoimmune disorders, and disrupted metabolic conditions, are now associated with particular aminoacyl-tRNA synthetases [31]. The metabolites linked to Aminoacyl-tRNA biosynthesis, including tryptophan, succinate, sarcosine, propionate, threonine, acetate, urocanate, uracil, isovalerate, and pyroglutamate, showed diverse expression patterns among seal species. While threonine, urocanate, uracil, and pyroglutamate exhibited varied expression patterns among seals, their levels may be associated with the role of the small intestinal mucosa in degrading certain amino acids in the diet [32–34]. The metabolite, urocanate, an intermediate in the histidine degradation pathway, has been identified as a molecule that stimulates bacterial infection [35]. Furthermore, desaminotyrosine, a metabolite associated with microbes and known for its protective role against influenza [36], was higher in *A. pusillus pusillus* in both the stomach and colon.

Acetate, succinate, lactate, butyrate, and other metabolites presented insights into the fermentation process and microbial activity in the gut. Acetate, a primary short-chain fatty acid, was upregulated in several comparisons. A distinctively elevated level of acetate, with a fold change exceeding 2, was observed in both colon and stomach samples of *A. pusillus pusillus* compared to other seals, indicating potential saccharolytic fermentation by intestinal microbiota [37]. A high percentage of acetate may also indicate an overgrowth of anaerobic flora, particularly Clostridium [38–40]. This suggests a profound influence of saccharolytic fermentation, possibly derived from the ingestion of indigestible carbohydrates, including those from algae species known for their high carbohydrate and lipid content [37]. The metabolite, succinate was upregulated in the colon content of *A. pusillus pusillus* versus *A. tropicalis* 2, *A. tropicalis* 1 versus *A. tropicalis* 2, *A. tropicalis* 1 versus *O. rossii*, and *A. tropicalis* 1 versus *L.*

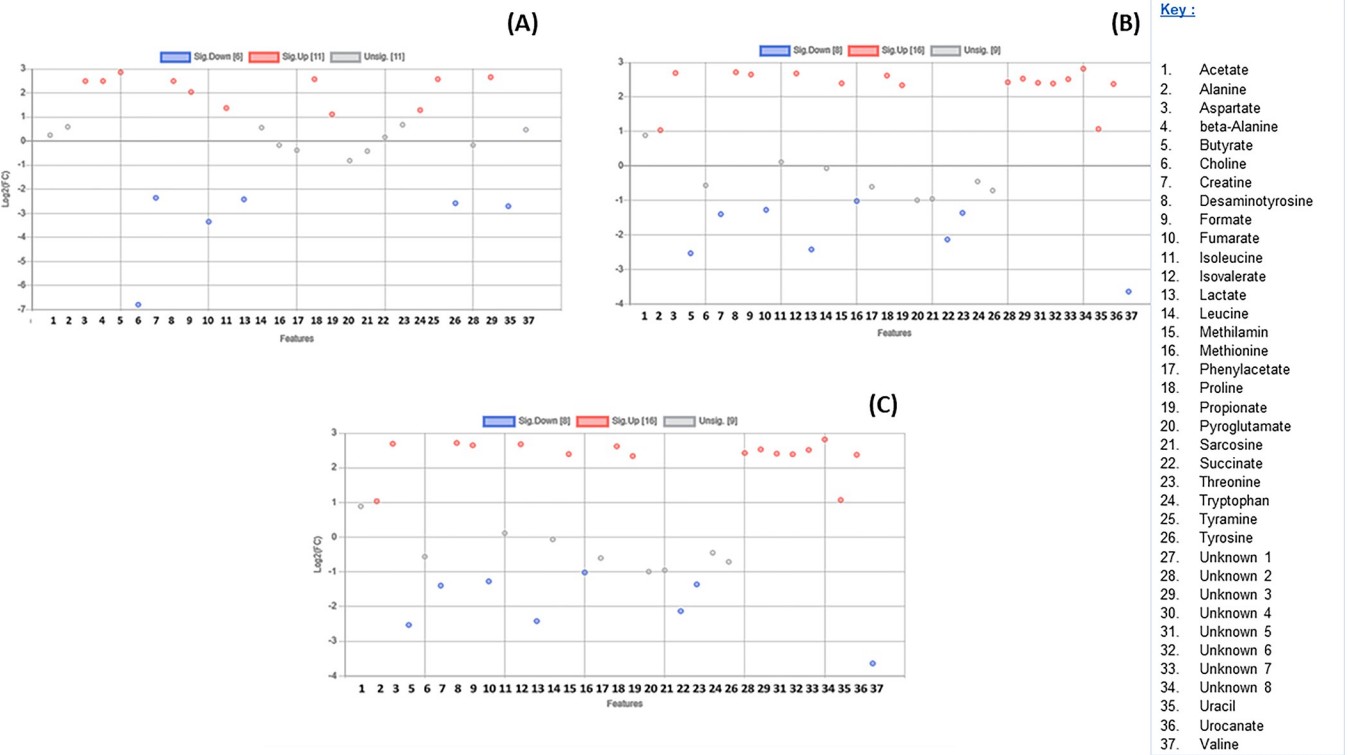

**Fig 9. A scatter plot illustrating the metabolites that are insignificant and significantly downregulated and upregulated in the colon of different seals.** (A) Metabolites in *O. rossii* versus *L. carcinophaga*, (B) Metabolites in *O. rossii* versus *A. pusillus pusillus* (C) Metabolites in *O. rossii* versus *A. tropicalis* 2.

*carcinophaga*. Succinate and lactate, intermediate metabolites derived from the fermentation process, showed varying concentrations dependent on diet and gut microbiota composition [41]. Lactate concentrations were consistently low in stomach and colon content comparisons, as lactate is swiftly transformed into propionate in the gut lumen [41,42]. In addition to acetate and propionate, the predominant short-chain fatty acid resulting from microbial nutrient breakdown is butyrate [43], significantly expressed higher in *A. pusillus pusillus*.

Branched-chain amino acids (valine, isoleucine, leucine) play a crucial role in brain function. The amino acids are generally downregulated in the seals, except in the upregulation comparison of colon contents of *O. rossii* versus *L. carcinophaga* and *A. tropicalis* 1 versus *A. tropicalis* 2. They are transferred alongside aromatic amino acids into the brain, influencing the synthesis of some neurotransmitters [44,45] and are associated with Alzheimer's-like conditions in these animals [46,47]. This study highlighted the potential transformation of branched-chain amino acids, including valine, leucine, and isoleucine into branched-chain short-chain fatty acids, such as isobutyrate and isovalerate [48]. These branched-chain short-chain fatty acids contribute minimally (5%) to the total production of short-chain fatty acids [48]. Isobutyrate and isovalerate were expressed at low levels in the colon and stomach contents of the seals.

Tryptophan, an essential amino acid is susceptible to stress, infections and fluctuations in the gut microbiome that can shift tryptophan metabolism from serotonin production towards the kynurenic pathway [49–52]. While it is significantly associated with Alzheimer's disease, tryptophan was not upregulated in all the samples tested. Thus, it may unlikely be associated with the death of the stranded seals. Thus, the presence of tryptophan in the seals may be

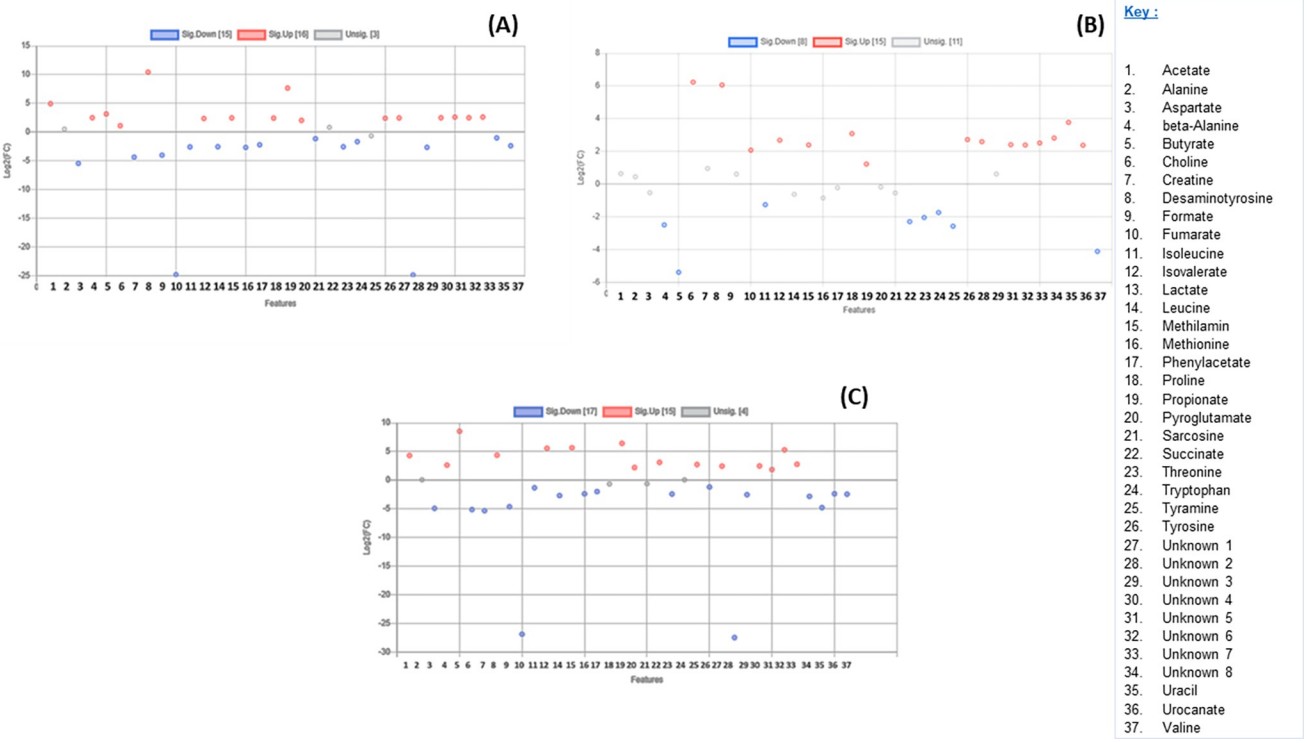

**Fig 10. A scatter plot illustrating the metabolites that are insignificant and significantly downregulated and upregulated in the colon of different seals.** (A) Metabolites in *L. carcinophaga*, versus *A. pusillus pusillus* (B) Metabolites in *L. carcinophaga*, versus *A. tropicalis* 2. (C) Metabolites in *A. pusillus pusillus* versus *A. tropicalis* 2.

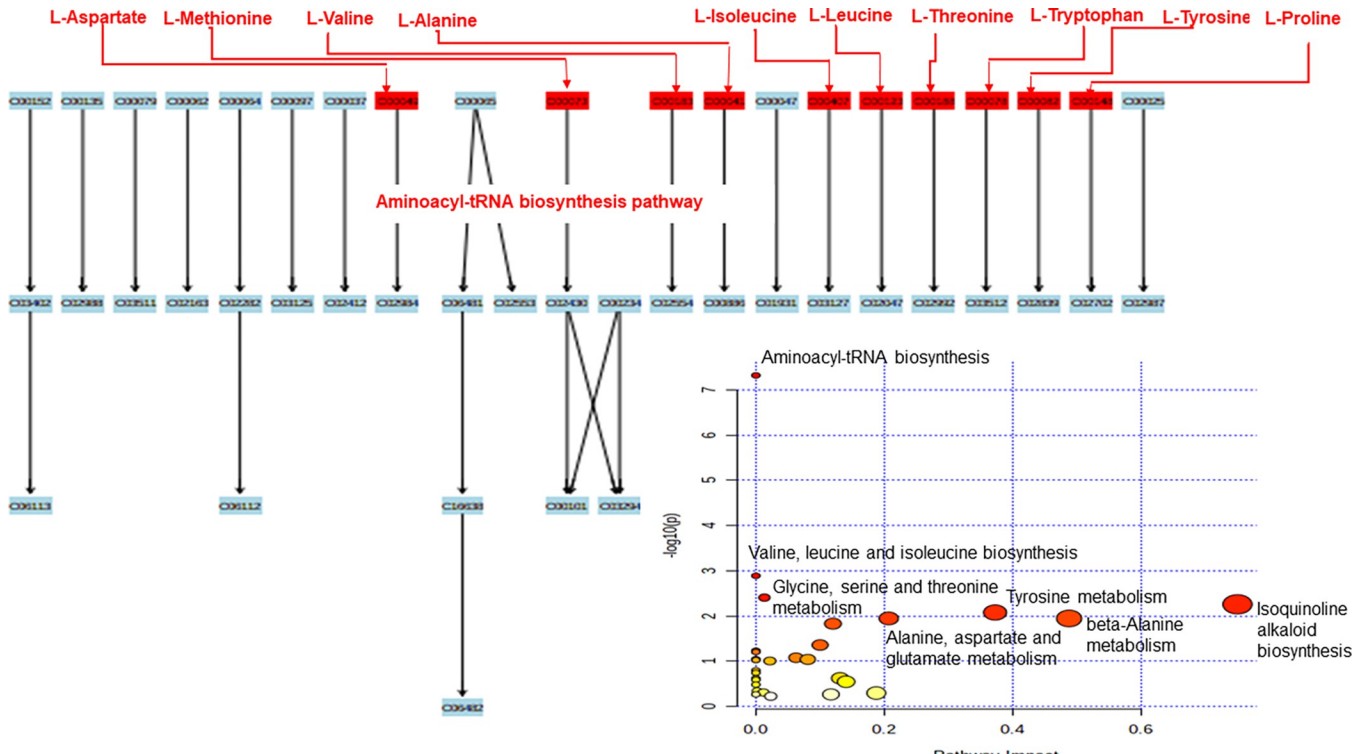

**Fig 11. The MetPA represents a summary of the most perturbed biochemical pathway analysis using enrichment analysis and topology analysis.** The enrichment analysis also showed the aminoacyl-tRNA biosynthesis as the most significantly enriched pathway.

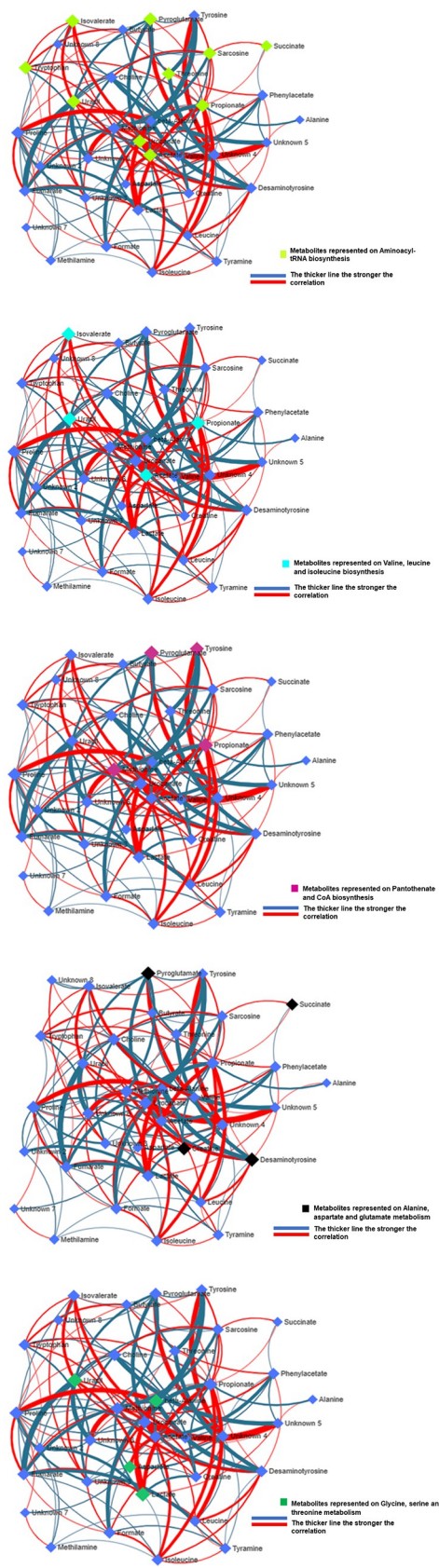

**Fig 12. A. Debiased Sparse Partial Correlation (DSPC) network model for estimating metabolic correlation networks that allows defining direct interactions between metabolites.** DSPC network model also depicted the metabolites that form the biochemical pathway of aminoacyl-tRNA biosynthesis. **B. Debiased Sparse Partial Correlation (DSPC) network model for estimating metabolic correlation networks that allows defining direct interactions between metabolites.** DSPC network model also depicted the metabolites that form the biochemical pathway of valine, leucine and isoleucine biosynthesis. **C. Debiased Sparse Partial Correlation (DSPC) network model for estimating metabolic correlation networks that allows defining direct interactions between metabolites.** DSPC network model also depicted the metabolites that form the biochemical pathway of pantothenate and CoA biosynthesis. **D. Debiased Sparse Partial Correlation (DSPC) network model for estimating metabolic correlation networks that allows defining direct interactions between metabolites.** DSPC network model also depicted the metabolites that form the biochemical pathway of alanine, aspartate and glutamate metabolism. **E. Debiased Sparse Partial Correlation (DSPC) network model for estimating metabolic correlation networks that allows defining direct interactions between metabolites.** DSPC network model also depicted the metabolites that form the biochemical pathway of glycine, serine and threonine metabolism.

**Table 2. Metabolites enriched pathways mapped to the Debiased Sparse Partial Correlation (DSPC) network to create a subnetwork.**

| Pathways | Total | Expected | Hits | P-values | Topology | PVal.Z | Topo.Z |
|---|---|---|---|---|---|---|---|
| Aminoacyl-tRNA biosynthesis | 48 | 0.805 | 10 | 1.12e-09 | 0.213 | 6.21 | 1.05 |
| Valine, leucine and isoleucine biosynthesis | 8 | 0.134 | 4 | 4.16e-06 | 0.571 | 3.56 | 3.67 |
| Pantothenate and CoA biosynthesis | 19 | 0.319 | 4 | 0.000203 | 0.278 | 2.3 | 1.52 |
| Alanine, aspartate and glutamate metabolism | 28 | 0.469 | 4 | 0.000968 | 0.444 | 1.8 | 2.74 |
| Glycine, serine and threonine metabolism | 33 | 0.553 | 4 | 0.00183 | 0.188 | 1.59 | 0.865 |
| beta-Alanine metabolism | 21 | 0.352 | 3 | 0.00456 | 0.6 | 1.3 | 3.88 |
| Pyruvate metabolism | 22 | 0.369 | 3 | 0.00522 | 0.286 | 1.25 | 1.58 |
| Propanoate metabolism | 23 | 0.386 | 3 | 0.00593 | 0.136 | 1.21 | 0.492 |
| Phenylalanine metabolism | 10 | 0.168 | 2 | 0.0112 | 0.222 | 1 | 1.12 |
| Arginine biosynthesis | 14 | 0.235 | 2 | 0.0217 | 0.154 | 0.79 | 0.62 |
| Butanoate metabolism | 15 | 0.251 | 2 | 0.0248 | 0.143 | 0.748 | 0.539 |
| Valine, leucine and isoleucine degradation | 40 | 0.671 | 3 | 0.0274 | 0.0769 | 0.716 | 0.058 |
| Histidine metabolism | 16 | 0.268 | 2 | 0.0281 | 0.2 | 0.708 | 0.957 |
| Tyrosine metabolism | 42 | 0.704 | 3 | 0.0311 | 0.22 | 0.674 | 1.1 |
| Citrate cycle (TCA cycle) | 20 | 0.335 | 2 | 0.0427 | 0.211 | 0.573 | 1.03 |

**Key**: PVal.Z = P value from Z score; Topo.Z = Topology from Z score.

attributed to the seals' gut microbiome or derived from other animals, possibly fish caught at sea [53].

Choline, an essential dietary precursor, was upregulated in the seals, particularly in *A. tropicalis* 1, suggesting potential dietary variations. Choline, occurring as phosphatidylcholine [54], plays a crucial role in various physiological functions [55,56]. The stomach and colon contents of *A. pusillus pusillus*, serving as a dietary precursor for gut microbial trimethylamine production, may contribute to cardiovascular risk factors [57,58]. Another metabolite, Sarcosine, formed from the metabolism of choline and methionine, is converted quickly to glycine and serves as a metabolic source for elements like creatine [59]. Although creatine was generally downregulated in seal species, it exhibited upregulation in the stomach content of *A. tropicalis* 1 versus *L. carcinophaga* and *A. tropicalis* 1 versus *A. tropicalis* 2, as well as in the colon content of *A. tropicalis* 1 versus *L. carcinophaga*. Creatine levels, essential for ATP energy delivery in the colon, are maintained through diet and endogenous synthesis from arginine and glycine. This energy supports the formation of an intact barrier in the colon, preventing the passage of

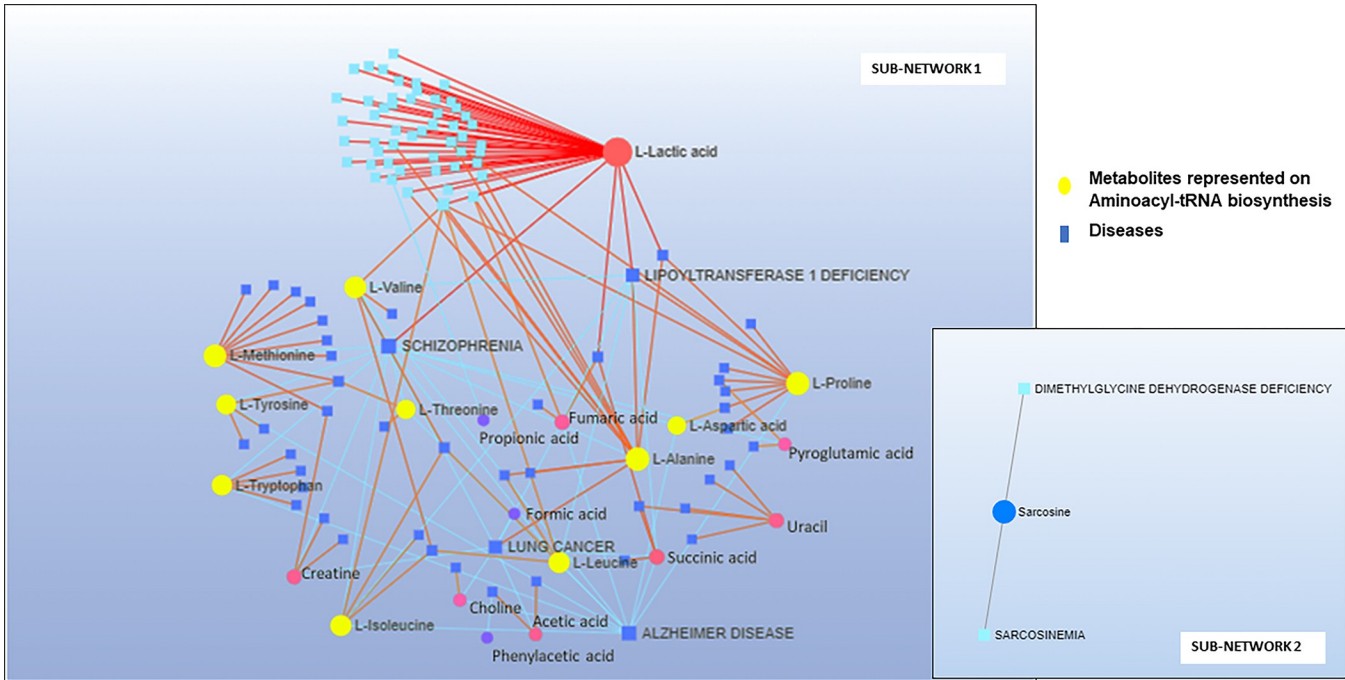

**Fig 13. A typical example of the metabolite and disease interaction networks of the samples from the stomach and colon of *A. pusillus pusillus*, *L. carcinophaga*, *O. rossii* and *A. tropicalis* 1 and 2 which provide a visualization of the diseases related metabolites identified from metabolomics.**

molecules and mitigating inflammation, as observed in conditions like ulcerative colitis [60,61].

In addition to producing short-chain fatty acids, the gut microbiota is responsible for generating other metabolites like methylamine and indoles. Methylamine, derived from dietary choline [62], was found to be higher in *A. pusillus pusillus* in both the stomach and colon. The hypothesis proposed by [63] regarding the connection between colon metabolites, diet, and microbiota composition is relevant to this study [63]. Variability in metabolite quantities between seal species may be attributed to differences in gut microbiota rather than dietary intake. While the dominant gut microbiota phyla in seals are Firmicutes, Fusobacteria, Proteobacteria, and Bacteroidetes [64], direct evaluation of the gut microbiome in the samples was not possible. Lack of data on blood serum metabolite quantities further limits the interpretation of absorbed metabolites. Future experimental investigations are needed to elucidate the biological consequences of these pathway networks, particularly regarding seal dietary exposure and potential causes of death.

Although the exact cause of death in stranded seals is unclear, the presence of certain metabolites in the stomach and colon suggests a possible origin from the gut microbiome or other marine organisms, likely fish. Algae, known for high concentrations of specific metabolites [65–67], form the foundation of the aquatic food chain, providing food for fish, which, in turn, are prey for seals [68,69]. While the nonspecific nature of results is acknowledged, the limitations inherent to metabolomics analyses should be considered. There is a need for specific and reliable biomarkers indicating dietary exposures across various seal populations, along with relevant metabolite and disease interaction networks to explore disease-related metabolites in seals. Despite these challenges, our metabolomic approach provides insights into potential metabolites in the stomach and colon of the seals.

## Conclusion

The metabolomic analysis of stomach and colon contents from the seals provides valuable insights into the complex interplay between diet, gut microbiota, and metabolite profiles. The variations observed highlight the need for further investigations to elucidate the biological consequences of these pathway networks and their relevance to seal dietary exposure and potential causes of death. While the exact cause of death remains unclear, the presence of specific metabolites may offer signs to dietary habits and microbial interactions in these marine mammals. Future studies incorporating gut microbiome analysis, clinical phenotypes, and blood serum metabolomics will contribute to a more comprehensive understanding of the intricate connections between diet, gut health, and overall metabolic health in seals.

## Supporting information

**S1 File. Typical examples of the metabolite's spectra from the stomach and colon samples of** *Arctocephalus pusillus pusillus, Arctocephalus tropicalis, Lobodon carcinophaga and Ommatophoca rossii.*
(PDF)

## Acknowledgments

The authors are grateful to National Zoological Garden (NZG) forensic laboratory Pretoria for allowing us to conduct our studies at their facility. We also like to thank Bayworld Museum for their assistance in terms of sample collections.

## Author Contributions

**Conceptualization:** Mukhethwa Micheal Mphephu, Monica Mwale, Nqobile Monate Mkolo.

**Data curation:** Mukhethwa Micheal Mphephu, Greg Hofmeyer, Nqobile Monate Mkolo.

**Formal analysis:** Mukhethwa Micheal Mphephu, Oyinlola Oluwunmi Olaokun, Caswell Mavimbela, Nqobile Monate Mkolo.

**Funding acquisition:** Mukhethwa Micheal Mphephu.

**Investigation:** Mukhethwa Micheal Mphephu.

**Methodology:** Mukhethwa Micheal Mphephu, Greg Hofmeyer, Nqobile Monate Mkolo.

**Project administration:** Mukhethwa Micheal Mphephu, Nqobile Monate Mkolo.

**Resources:** Caswell Mavimbela, Greg Hofmeyer, Nqobile Monate Mkolo.

**Supervision:** Oyinlola Oluwunmi Olaokun, Caswell Mavimbela, Monica Mwale, Nqobile Monate Mkolo.

**Validation:** Nqobile Monate Mkolo.

**Visualization:** Mukhethwa Micheal Mphephu, Nqobile Monate Mkolo.

**Writing – original draft:** Mukhethwa Micheal Mphephu.

**Writing – review & editing:** Oyinlola Oluwunmi Olaokun, Caswell Mavimbela, Monica Mwale, Nqobile Monate Mkolo.

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
