## [Decision Letter · Decision Letter 0]

6 Sep 2023

PONE-D-23-21408Metabolomics approach for stomach and colon contents analysis in Arctocephalus pusillus, Arctocephalus tropicalis, Lobodon carcinophaga and Ommatophoca rossii from Sub-Antarctic regionPLOS ONE

Dear Dr. MKOLO,

Thank you for submitting your manuscript to PLOS ONE. After careful consideration, we feel that it has merit but does not fully meet PLOS ONE’s publication criteria as it currently stands. Therefore, we invite you to submit a revised version of the manuscript that addresses the points raised during the review process.

We look forward to receiving your revised manuscript.

Kind regards,

Ishtiyaq Ahmad, Ph.D

Academic Editor

PLOS ONE

Journal Requirements:

https://www.nature.com/articles/s41419-019-2145-5?code=af4a8ff1-2bc0-4b3a-bd27-362dd25afcbb&error=cookies_not_supported

https://www.ncbi.nlm.nih.gov/pmc/articles/PMC7468900/

https://www.researchgate.net/publication/332275176_Tryptophan_Metabolic_Pathways_and_Brain_Serotonergic_Activity_A_Comparative_Review

https://www.mdpi.com/1420-3049/25/21/5101/xml

In your revision ensure you cite all your sources (including your own works), and quote or rephrase any duplicated text outside the methods section. Further consideration is dependent on these concerns being addressed.

Reviewers' comments:

Reviewer's Responses to Questions

**Comments to the Author**

1. Is the manuscript technically sound, and do the data support the conclusions?

Reviewer #1: Yes

2. Has the statistical analysis been performed appropriately and rigorously? 

Reviewer #1: Yes

3. Have the authors made all data underlying the findings in their manuscript fully available?

Reviewer #1: Yes

4. Is the manuscript presented in an intelligible fashion and written in standard English?

Reviewer #1: Yes

5. Review Comments to the Author

Reviewer #1: The aim of this review was to evaluate ‘Metabolomics approach for stomach and colon contents analysis in Arctocephalus pusillus, Arctocephalus tropicalis, Lobodon carcinophaga and Ommatophoca rossii from Sub-Antarctic region’. Thus, this manuscript represents a valid contribution to the understanding of the importance of using metabolomics technique for the examination of stomach and gut contents. Based on that, I consider the manuscript with interest to be published but with major revision. Nevertheless, before it can be published in its final form, I have some concerns that need to be addressed. There is repetition of sentences at some places and some of the sentences seem meaningless. Mostly old references have been cited. So, overall the manuscript needs an exhaustive revision before it can be considered for publication.

Title:

The title does not convey the significance or implications of the study's findings and making it difficult to grasp the focus of the study.

Abstract:

Abstract may be unclear to readers without a specific background in the subject. It is of concern that the abstract could have been catchier about the importance of study. Overall, the results aren’t reflected in a comprehensive manner and don’t signify the importance of study.

Contributions to the field:

The contributions to the field statement could be improved. I suggest that this section be rewritten. Moreover, authors didn’t provide the choice of selection of the species. I think it is a good idea to conclude by summarizing the practical implications of the study's findings.

Introduction:

The introduction is quite short (for the information it aims to provide) and could benefit from being more elaborative providing the significance of study in reference to relevant literature. Most significantly, the introduction also lacks a clear thesis statement that succinctly summarizes the main objectives and purpose of the study.

Materials and Methods:

I find this section to be generally well-written. It provides sufficient details on the study area, sampling, identification etc.

Results:

I find the results section to be comprehensive and detailed.

Discussion and conclusion:

Again, I find this section to be comprehensive and detailed. But some sentences are unnecessarily long and complex, which makes the text difficult to read; break them up for clarity. Old references must be replaced with latest ones. Conclusion hasn’t been well synthesized lacking coherence with the focus and findings of the study.

6. PLOS authors have the option to publish the peer review history of their article (what does this mean?). If published, this will include your full peer review and any attached files.

Reviewer #1: No

---

## [Author Response · Author response to Decision Letter 0]

1 Feb 2024

Response to reviewers’ comments:

Rebuttal 

Reviewers’ comments and Response

1. The aim of this review was to evaluate ‘Metabolomics approach for stomach and colon contents analysis in Arctocephalus pusillus, Arctocephalus tropicalis, Lobodon carcinophaga and Ommatophoca rossii from Sub-Antarctic region’. Thus, this manuscript represents a valid contribution to the understanding of the importance of using metabolomics technique for the examination of stomach and gut contents. Based on that, I consider the manuscript with interest to be published but with major revision. Nevertheless, before it can be published in its final form, I have some concerns that need to be addressed. There is repetition of sentences at some places and some of the sentences seem meaningless. Mostly old references have been cited. So, overall the manuscript needs an exhaustive revision before it can be considered for publication. 

Thank you for the comment exhaustive revision have been done in paper. Sentences which are repeated were removed. Moreover, old references were noted and revised. 

2. Title:

The title does not convey the significance or implications of the study's findings and making it difficult to grasp the focus of the study.

The title was corrected. 

3. Abstract:

Abstract may be unclear to readers without a specific background in the subject. It is of concern that the abstract could have been catchier about the importance of study. Overall, the results aren’t reflected in a comprehensive manner and don’t signify the importance of study.

Contributions to the field:

The contributions to the field statement could be improved. I suggest that this section be rewritten. Moreover, authors didn’t provide the choice of selection of the species. I think it is a good idea to conclude by summarizing the practical implications of the study's findings. 

The abstract was rewritten and content on importance of the study and contribution to the field was added.

4. Introduction:

The introduction is quite short (for the information it aims to provide) and could benefit from being more elaborative providing the significance of study in reference to relevant literature. Most significantly, the introduction also lacks a clear thesis statement that succinctly summarizes the main objectives and purpose of the study.

The introduction was corrected and more information was provided.

5. Materials and Methods:

I find this section to be generally well-written. It provides sufficient details on the study area, sampling, identification etc. 

We thank you for the comment.

6. Results:

I find the results section to be comprehensive and detailed. 

We thank you for the comment.

7. Discussion and conclusion:

Again, I find this section to be comprehensive and detailed. But some sentences are unnecessarily long and complex, which makes the text difficult to read; break them up for clarity. Old references must be replaced with latest ones. Conclusion hasn’t been well synthesized lacking coherence with the focus and findings of the study. 

We thank you for the comment and this section was corrected accordingly.

---

## [Decision Letter · Decision Letter 1]

27 Feb 2024

Metabolomics approach for predicting stomach and colon contents in dead Arctocephalus pusillus, Arctocephalus tropicalis, Lobodon carcinophaga and Ommatophoca rossii from Sub-Antarctic region

PONE-D-23-21408R1

Dear Dr. Mkolo,

We’re pleased to inform you that your manuscript has been judged scientifically suitable for publication and will be formally accepted for publication once it meets all outstanding technical requirements.

Kind regards,

Ishtiyaq Ahmad, Ph.D

Academic Editor

PLOS ONE

Additional Editor Comments (optional):

Authors have adressed all the queries as suggested by the referee. Hence, needs no further improvement. Paper is accepted for publication.

Reviewers' comments:

Reviewer's Responses to Questions

**Comments to the Author**

1. If the authors have adequately addressed your comments raised in a previous round of review and you feel that this manuscript is now acceptable for publication, you may indicate that here to bypass the “Comments to the Author” section, enter your conflict of interest statement in the “Confidential to Editor” section, and submit your "Accept" recommendation.

Reviewer #1: All comments have been addressed

2. Is the manuscript technically sound, and do the data support the conclusions?

Reviewer #1: Yes

3. Has the statistical analysis been performed appropriately and rigorously? 

Reviewer #1: Yes

4. Have the authors made all data underlying the findings in their manuscript fully available?

Reviewer #1: Yes

5. Is the manuscript presented in an intelligible fashion and written in standard English?

Reviewer #1: Yes

6. Review Comments to the Author

Reviewer #1: (No Response)

7. PLOS authors have the option to publish the peer review history of their article (what does this mean?). If published, this will include your full peer review and any attached files.

Reviewer #1: **Yes: **Dr. Kousar Jan

---

## [Editor Report · Acceptance letter]

21 Mar 2024

PONE-D-23-21408R1 

PLOS ONE

Dear Dr. Mkolo, 

I'm pleased to inform you that your manuscript has been deemed suitable for publication in PLOS ONE. Congratulations! Your manuscript is now being handed over to our production team.

Kind regards, 

on behalf of

Dr. Ishtiyaq Ahmad 

Academic Editor

PLOS ONE